# Phosphomimicry on STAU1 Serine 20 Impairs STAU1 Posttranscriptional Functions and Induces Apoptosis in Human Transformed Cells

**DOI:** 10.3390/ijms23137344

**Published:** 2022-07-01

**Authors:** Yulemi Gonzalez Quesada, Florence Bonnet-Magnaval, Luc DesGroseillers

**Affiliations:** Département de Biochimie et Médecine Moléculaire, Faculté de Médecine, Université de Montréal, 2900 Édouard Montpetit, Montreal, QC H3T 1J4, Canada; yulemi.gonzalez.quesada@umontreal.ca (Y.G.Q.); florence.bonnet@umontreal.ca (F.B.-M.)

**Keywords:** Staufen 1, apoptosis, cell proliferation, posttranscriptional regulation

## Abstract

Staufen 1 (STAU1) is an RNA-binding protein that is essential in untransformed cells. In cancer cells, it is rather STAU1 overexpression that impairs cell proliferation. In this paper, we show that a modest increase in STAU1 expression in cancer cells triggers apoptosis as early as 12 h post-transfection and impairs proliferation in non-apoptotic cells for several days. Interestingly, a mutation that mimics the phosphorylation of STAU1 serine 20 is sufficient to cause these phenotypes, indicating that serine 20 is at the heart of the molecular mechanism leading to apoptosis. Mechanistically, phosphomimicry on serine 20 alters the ability of STAU1 to regulate translation and the decay of STAU1-bound mRNAs, indicating that the posttranscriptional regulation of mRNAs by STAU1 controls the balance between proliferation and apoptosis. Unexpectedly, the expression of RBD2^S20D^, the N-terminal 88 amino acids with no RNA-binding activity, is sufficient to induce apoptosis via alteration, in trans, of the posttranscriptional functions of endogenous STAU1. These results suggest that STAU1 is a sensor that controls the balance between cell proliferation and apoptosis, and, therefore, may be considered as a novel therapeutic target against cancer.

## 1. Introduction

Cell proliferation is a complex phenomenon that relies on the tight regulation of gene expression, signaling pathways, and proteome balance [1,2,3,4]. Cell survival and adaptation depend on the capacity of the cells to rapidly respond to several internal and external stresses that are continuously challenging their homeostasis. The dysregulation of any of these pathways may trigger severe abnormalities that cause diseases or induce cell death [4,5]. For example, the activation of oncogenes is well known to stimulate cell proliferation and thus induce tumorigenesis in many cancer types [6]. However, oncogenic mutations or the overexpression of oncogenes can rather induce cellular senescence [7]. Similarly, cancer cells use a variety of molecular mechanisms to suppress apoptosis and facilitate cell proliferation [8,9]. Thus, cell proliferation, senescence, and apoptosis form intricate pathways, in which even the subtle dysregulation of a gene or protein expression can tip the balance toward one or the other cell decision. Understanding the molecular mechanisms beyond the cell decision may provide deeper insights into cancer and influence therapeutic strategy.

STAU1 is expressed as two different isoforms of 55 and 63 kDa [10]. STAU1^63^ is identical to STAU1^55^ but carries an 81-amino acid extension at its N-terminal extremity. STAU1^55^ is a double-stranded RNA-binding protein that plays critical roles in cell decisions via the posttranscriptional regulation of RNA regulons [10,11,12,13]. STAU1^55^ binds to and controls the expression of different populations of specific mRNAs [14,15,16,17,18] through the induction or inhibition of their transport and localization [19,20], alternative splicing [21], translation [15,17,18,22], or decay [23]. STAU1^55^ is associated with ribosomes and regulates cap-dependent translation [10,11,24]. It also acts as an IRES-transacting factor in cap-independent translation [25]. Through these molecular mechanisms, STAU1^55^ regulates several physiological pathways linked to cell decision such as differentiation, proliferation, migration, apoptosis, autophagy, and stress response autophagy (reviewed in [13,19,26,27,28,29]). In cancer cells, the proto-oncogene non-receptor tyrosine kinase SRC phosphorylates STAU1^55^ on tyrosines 380 and 493 [30]. Although the meaning of these modifications is not known, it suggests that posttranslational modifications could modify the molecular functions of STAU1^55^ in cancer cells compared to those in untransformed cells.

STAU1^55^ expression is essential for the proliferation of non-transformed cells, as it facilitates the cell cycle checkpoint transition via a coordinated control of pro/anti-proliferative and pro/anti-apoptotic regulons [31]. Not surprisingly then, large scale comparative studies of tumor and normal tissues indicate that the STAU1^55^ expression level is upregulated in most cancers, favoring the pro-proliferative (reviewed in [13,29]) and malignant [32,33] phenotypes. It was suggested that STAU1^55^ could be considered as an oncogene and that the high level of STAU1^55^ could indeed be a potential biomarker for prostate cancer [34], high grade gliomas [35,36], and stage IA and IB lung squamous cell carcinoma [37]. Thus, by adjusting the expression of STAU1^55^, the cancer cell establishes a new balance of gene expression via the STAU1^55^-mediated posttranscriptional regulation that favors both cell proliferation and survival. However, genome-wide studies of large cohorts of tumors also show a correlation between high STAU1^55^ level and long-term survival rates following treatments [38,39,40]. These opposite results suggest that STAU1^55^ expression marks the boundary between proliferation and cell death and that a small modulation of its expression could tip the balance towards one of the two pathways. Indeed, ectopic expression of STAU1^55^ in cancer cells triggers signaling pathways that lead to mitotic defects [41] and increases sensitivity to apoptosis induction [42,43,44]. Similarly, proliferation is facilitated and apoptosis is reduced when STAU1^55^-mediated RNA decay (SMD) is inhibited by the upregulation of long non-coding RNAs in several tumors [33,42,43,45], indicating that SMD controls cell decision.

STAU1^55^ expression is transcriptionally upregulated by the transcription factor E2F1 [31] and proteolytically downregulated by the E3-ubiquitin ligase anaphase-promoting complex/cyclosome (APC/C) during mitosis [41]. However, the mechanisms that control its functions in proliferation versus apoptosis are not clear. We previously showed that a modest increase in STAU1^55^ levels impairs cancer cell proliferation and that the molecular determinant involved in this antiproliferative effect is located in the first N-terminal 88 amino acids of STAU1^55^ [41]. This sequence corresponds to STAU1^55^ RBD2, a domain that does not bind RNAs in vitro but establishes protein–protein interaction with CDH1 and CDC20 for its degradation in mitosis by APC/C [41] and with the mitotic spindle [46]. In this paper, we show that STAU1^55^ overexpression in transformed cells induces apoptosis and proliferation impairment dependent on a molecular determinant located within amino acid 17 and 25 at the N-terminal end of STAU1^55^. More specifically, the presence of a negative charge on S20 is at the heart of the mechanism that induces apoptosis and proliferation impairment in cancer upon STAU1^55^ overexpression via altered STAU1^55^ posttranscriptional functions.

## 2. Results

### 2.1. STAU1^55^ Overexpression Causes a Fast-Acting Apoptosis Response in Transformed Cells

Our previous results demonstrated that an increase in the level of STAU1^55^ expression is deleterious to cancer cells [41]. To determine if this phenotype is due to apoptosis, we performed time-lapse experiments using a coupled-GFP dye that specifically stains activated caspases 3/7 in cellulo, thus allowing a real-time detection of apoptosis induction. First, HEK293T cells were transfected with plasmids coding for STAU1^55^-mCherry or mCherry as the control. The GFP-dye was directly added after transfection, and the time-lapse microscopy acquisition was performed for 36 h (Figure 1A). A significant increment of GFP staining (apoptosis) was observed 12 h after the transfection in STAU1^55^-mCherry transfected cells but not in control cells (Figure 1B,C). At this time point, mCherry staining (STAU1^55^ and control expression) was barely detectable in both cell lines (Figure 1B,D). Nevertheless, the western blotting analysis showed that STAU1^55^-mCherry and the control mCherry were expressed at this time point (Figure 1D). The GFP signal intensity then reached a maximum value at around 18–24 h, and then stabilized until 36 h (Figure 1B,C). Simultaneously, the mCherry signal slowly increased until the end of the experiment (Figure 1B,D). This result revealed that STAU1^55^ overexpression triggers cell apoptosis during the first hours following transfection. Interestingly, although the induction of apoptosis correlates with STAU1^55^ expression, STAU1^55^ overexpression (red signal) was not detected at this time point, suggesting that the cell decision to activate the apoptotic cascade occurs in the early steps following STAU1^55^ expression and is not due to the excessive or artificial overexpression of STAU1^55^. 

The stabilization of the apoptotic GFP signal suggests that once the cell fate is decided, STAU1^55^-mCherry-expressing cells no longer enter apoptosis. To test this hypothesis, the addition of the GFP-dye and time-lapse microscopy was initiated 24 h post-transfection (Figure 1A), during the apoptotic plateau (Figure 1C). Prior to adding the GFP-dye, cells were washed to remove dead cells. After washing, the amount of apoptotic green signal was similar in the control and STAU1^55^-mCherry-expressing cells (Figure 1E). Images were then acquired for 12 h. Interestingly, in contrast to what was observed in the first 24 h, there was no significant difference between the two cell lines in the apoptotic GFP signal (Figure 1F), indicating that apoptosis is an early response to STAU1^55^ expression. 

### 2.2. Amino Acids 18–25 Carry the Molecular Determinant That Impairs Cell Proliferation

To determine the fate of non-apoptotic cells, growth curves were generated. Controls (empty vector and STAU1^Δ88^-HA_3_) and STAU1^55^-HA_3_-transfected cells were trypsinized 24 h post-transfection to remove apoptotic cells, then plated at the same density, and allowed to grow for three days (Figure 2). As previously observed [41], STAU1^55^-expressing cells had impaired cell proliferation profiles compared to control cells, indicating that cells that did not enter the apoptotic pathway nevertheless have proliferation defects.

To identify the molecular determinant within the fist 88 amino acids at the N-terminal of STAU1^55^ that impairs cell proliferation, we generated progressive deletions in this region (Figure 2A). The wild type and mutant proteins were expressed in HEK293T cells (Figure 2B), and growth curve assays were initiated 24 h post-transfection (Figure 2C) to determine the capacity of each mutant to impair cell proliferation. As controls, the expression of STAU1^55^-HA_3_ impaired cell proliferation, whereas the mutant protein with deletion of the first 88 N-terminal amino acids (STAU1^Δ88^-HA_3_) grew as efficiently as untransfected or empty vector-transfected cells. The expression of mutants that lacked the first 17 N-terminal amino acids (or less) still impaired cell proliferation, as did the wild-type protein. In contrast, the expression of proteins with the deletion of 25 (or more) amino acids did not impair cell proliferation. These results indicate that the molecular determinant responsible for the antiproliferative effect of STAU1^55^ is comprised between amino acid 18 and 25 at the N-terminal extremity of STAU1^55^.

### 2.3. Phosphomimicry on S20 and T21 Controls Cell Proliferation

Remarkably, four of the eight amino acids in the 18–25 amino acid sequence were putative targets for phosphorylation, MQS^20^T^21^Y^22^NY^24^N. To determine if phosphorylation events could be involved in the STAU1^55^-dependent impairment of cell proliferation, we generated phosphomimetic and non-phosphorylatable mutants for each of these residues and determined the impact of these modifications on cell proliferation using the growth curve assay (Figure 3). Each mutant was transfected in HEK293T cells, and western blotting was performed to confirm their expression (Figure 3A). The expression of the phosphomimetic S20D mutant impaired cell proliferation (Figure 3B). In contrast, cells transfected with the non-phosphorylatable S20A mutant grew normally. The opposite effect was observed following the expression of the T21 mutants (Figure 3C). The non-phosphorylatable T21A mutant impaired cell proliferation, as did the wild-type protein. Cells expressing the phosphomimetic T21D mutant grew even faster than the control cells. These results indicate that the presence of a permanent negative charge on S20 affects cell proliferation and suggest that phosphorylation may account for the antiproliferative effect. In contrast, no differences in cell growth were observed when the phosphomimetic and non-phosphorylatable mutants of Y22 and Y24 were expressed (Figure 3D,E). 

To determine if STAU1^S20D^ also recapitulates the apoptotic program induced by STAU1^55^ when expressed in transformed cells, we transfected plasmids coding for STAU1^55^HA_3_, STAU1^S20A^-HA_3_, and STAU1^S20D^-HA_3_ in HEK293T cells and measured the fluorescence generated by the caspase activation. STAU1^S20D^-HA_3_ induced apoptosis, as did STAU1^55^-HA_3_ (Figure 3F). In contrast, STAU1^S20A^-HA_3_ had a limited capacity to induce apoptosis. A western blot experiment indicated that the proteins were expressed at the same levels (Figure 3G). These results indicate that the presence of a negative charge on S20 induces apoptosis and impairs the cell proliferation of non-apoptotic cells.

### 2.4. S20 Phosphomimicry Controls STAU1^55^-Mediated Post-Transcriptional Regulation

To identify the mechanism that impairs cell proliferation when STAU1^55^ is expressed in transformed cells, we determined whether the presence or absence of a negative charge on S20 altered STAU1^55^ posttranscriptional functions. Therefore, STAU1^55^, STAU1^S20A^, STAU1^S20D^, and the controls were expressed in HEK293T cells, and their molecular functions were compared. Our results indicate that the sub-cellular localization of the proteins in the cytoplasm or on the mitotic spindle and their stability/degradation were not affected by the mutations (data not shown).

In contrast, the capacity of STAU1^55^ to enhance translation when bound to the 5′UTR of mRNAs [22] was affected by the mutation on S20. We used the reporters Rluc fused (SBS-Rluc) or not (Rluc) to the ARF1 STAU1^55^-binding site (SBS) in the 5′UTR [22] to measure the impact of S20 mutations on translation. Plasmids coding for Rluc and SBS-Rluc were co-transfected with plasmids coding for STAU1^55^-YFP, STAU1^S20A^-YFP, STAU1^S20D^-YFP, or the empty vector. Luciferase assays were performed as a measure of translation. Western blotting experiments indicated that the levels of expression of STAU1^55^ mutants were similar and that their expression was equivalent to that of endogenous STAU1^55^ (Figure 4A). Luciferase assays first indicated, as expected [22], that the expression of STAU1^55^ enhanced the translation of SBS-Rluc compared to Rluc (Figure 4A). Then, they showed that STAU1^S20D^ was not able to enhance the translation of SBS-Rluc, whereas STAU1^S20A^ was still able. None of these proteins had a significant effect on the translation of Rluc.

Similarly, the capacity of STAU1^55^ to elicit SMD was affected by the S20D mutation. To measure SMD, we compared the amounts of ARF1 mRNA, a known target of SMD [23], in HCT116 and STAU1-KO HCT116 (CR1.3 cells) cells. As expected, the amounts of ARF1 mRNA were lower in HCT116 cells compared to those in STAU1-KO cells (Figure 4B). We then transfected STAU1-KO cells with plasmids coding for STAU1^55^-HA_3_, STAU1^S20A^-HA_3_, or STAU1^S20D^-HA_3_ and determined if SMD can be rescued. As expected, the expression of STAU1^55^-HA_3_ restored SMD in STAU1-KO cells, while the transfection of the empty vector had no effect. Interestingly, STAU1^S20A^-HA_3_ also rescued SMD, as did the STAU1^55^ wild type (Figure 4B). In contrast, STAU1^S20D^-HA_3_ was unable to rescue SMD. Altogether, these results indicate that the presence of a negative charge on S20 abrogates STAU1^55^-dependent translation and SMD.

### 2.5. RBD2 Expression Is Sufficient to Impair Cell Proliferation and to Induce Apoptosis

To determine if RBD2 is sufficient to impair cell proliferation and as corollary if other functional domains of STAU1^55^ are synergistically involved in this function, we fused RBD2 to YFP (Figure 5A), a protein that does not impair cell proliferation when expressed in transformed cells. Growth curve assays, initiated 24 h post-transfection, were used to compare the growth of cells expressing RBD2-YFP to that of cells expressing STAU1^55^-YFP, STAU1^Δ88^-YFP, and YFP, as controls (Figure 5B). Western blotting compared the expression of these proteins in HEK293T cells. As expected, the growth of cells expressing STAU1^55^-YFP was lower than those of cells expressing YFP or STAU1^Δ88^-YFP. Interestingly, cells expressing RBD2-YFP showed impaired cell proliferation, similar to that of cells expressing STAU1^55^-YFP. 

To determine if RBD2-YFP is able to induce apoptosis, cells were then transfected with plasmids coding for RBD2-YFP, STAU1^55^-YFP, or STAU1^Δ88^-YFP and immediately visualized by microscopy for fluorescence staining following caspase activation (Figure 5C). The expression of RBD2-YFP induced apoptosis, as did STAU1^55^-YFP, whereas STAU1^Δ88^-YFP did not. These results indicate that RBD2 is sufficient to impair cell proliferation and induce apoptosis when expressed in transformed cells and, consequently, that other domains of STAU1^55^ are not required for these functions. 

To determine if RBD2-YFP uses that same molecular determinant as STAU1^55^ to impair cell proliferation, we generated phosphomimetic and non-phosphorylatable mutants of S20 in the context of RBD2-YFP. The growth of cells expressing RBD2^S20A^-YFP and RBD2^S20D^-YFP was compared to that of cells expressing STAU1^55^-YFP, STAU1^Δ88^-YFP, RBD2-YFP, and YFP (Figure 5D). Interestingly, cells expressing RBD2^S20D^ showed impaired cell proliferation, as did STAU1^55^-YFP and RBD2-YFP, whereas cells expressing RBD2^S20A^-YFP grew normally, as did STAU1^Δ88^-YFP and YFP-expressing cells. These results indicate that the molecular determinant of RBD2-YFP that contributes to impair cell proliferation when expressed in transformed cells is identical to that of STAU1^55^. 

### 2.6. RBD2 Interferes with Endogenous STAU1^55^ to Impair SMD 

We showed that RBD2 impairs cell proliferation, as does STAU1^55^, and that it uses the same molecular determinant. However, RBD2 does not bind dsRNA. Therefore, we tested the capacity of RBD2 to affect STAU1^55^-dependent translation and SMD. Plasmids coding for Rluc and SBS-Rluc were co-transfected with plasmids coding for STAU1^55^-YFP, STAU1^Δ88^-YFP, RBD2-YFP, or the empty vector. Luciferase assays indicated that RBD2 expression did not enhance the translation of SBS-Rluc (Figure 6A). We then compared the effects of RBD2 on SMD. STAU1^55^-HA_3_, STAU1^Δ88^-HA_3_, RBD2-HA_3_, or the empty vector were transfected into STAU1-KO HCT116 cells. As expected, the expression of STAU1^55^-HA_3_ restored SMD, while STAU1^Δ88^-HA_3_ partly restored SMD. In contrast, the expression of RBD2-HA_3_ did not re-establish SMD (Figure 6B). 

Experiments in Figure 6A,B were designed to test the direct role of RBD2 in post-transcriptional regulation. As RBD2 had no direct role on translation or SMD, we then tested whether RBD2 interferes with endogenous STAU1^55^ and indirectly affects STAU1^55^-mediated posttranscriptional control. Especially, RBD2 was previously shown to interact with RBD5 [47], the C-terminal domain involved in STAU1^55^ homodimerization [48]. The expression of RBD2 could, therefore, prevent STAU1^55^ dimerization that in turn is essential for SMD [48]. Thus, HCT116 cells were transfected with increasing amounts of a plasmid coding for RBD2-HA_3_ (Figure 6C), and SMD efficiency was quantified with ARF1 mRNA (Figure 6D). STAU1-KO cells were used as the control for the amount of ARF1 mRNA when STAU1^55^ was absent. As expected, ARF1 mRNA levels were reduced in HCT116 cells compared to STAU1-KO cells due to SMD. The increasing expression of RBD2-HA_3_ paralleled an increase of ARF1 mRNA levels in STAU1^55^-expressing cells, consistent with impaired SMD. Combined with the fact that RBD2 has no effect on SMD in STAU1-KO cells (Figure 6B), these results indicate that RBD2 acts in *trans* and interferes with endogenous STAU1^55^ functions to prevent SMD.

## 3. Discussion

STAU1^55^ is well characterized for its involvement in cell decision during development, cell differentiation, or proliferation, through pluripotent post-transcriptional activities [13,29]. We now show that the posttranslational modification on serine 20 in the N-terminal region of STAU1^55^ is sufficient to impair cell proliferation and trigger the apoptosis of cancer cells. Indeed, the presence of a negative charge on serine 20 rather than overexpression, per se, is responsible for the observed phenotypes since the overexpression of STAU1^S20A^ has no effect on cell proliferation. The molecular mechanism by which serine 20 impairs cell proliferation likely relies on a modulation of STAU1^55^ posttranscriptional activity, especially the translation and/or decay of STAU1^55^-bound mRNAs. Surprisingly, expression of RBD2 alone is also able to impair cell proliferation dependent on serine 20, likely via an interaction that inhibits endogenous STAU1^55^ functions.

### 3.1. STAU1^55^ Overexpression Impairs Cell Proliferation and Triggers Apoptosis of Transformed Cells

In contrast to what was observed in untransformed cells [31], STAU1 depletion or knockout had no effect on the proliferation of cancer cells, indicating that STAU1 is not essential once cells are transformed [40,41,46]. In contrast, STAU1^55^ overexpression led to impaired cell proliferation that is due, at least partly, to the induction of cell death [41,42,43,44]. STAU1^55^ is described as an oncogene that facilitates cell cycle phase transition [31]. It is thus appropriate that its expression is upregulated in most cancers. However, STAU1^55^ expression seems to be kept at the edge between cell proliferation and cell death since high STAU1^55^ expression correlates with better survival rates following treatments [38,39,40], indicating that high STAU1^55^ expression makes cells more sensitive to death-inducing treatments. Therefore, an increase of STAU1^55^ expression in cancer cells perturbs the fragile equilibrium between proliferation and cell death and impairs cell proliferation. In addition to being an oncogene, STAU1^55^ can also be considered a pro-apoptotic factor. 

Interestingly, STAU1^55^-overexpressing transformed cells entered apoptosis shortly after transfection, well before exogenous STAU1^55^ expression was elevated. Cells that escaped apoptosis during this early response did not enter apoptosis later on. They nevertheless displayed impaired cell proliferation compared to untransfected cells. The nature of the defects was not clear. We previously showed that these cells are not apoptotic, senescent, or quiescent [41]. As STAU1^55^ is a facilitator of the cell cycle phase transition in untransformed cells [31], we believe that STAU1^55^ overexpression in these cells may cause an unregulated acceleration of phase transition that eventually triggers genetic stress and, in turn, impaired cell proliferation. 

### 3.2. STAU1^55^ Regulates Cell Proliferation through Modifications of Serine 20/Threonine 21

The molecular dissection of the N-terminal region of STAU1^55^ allowed us to identify two phosphorylatable residues, serine 20 and threonine 21, that exert opposite effects on cell proliferation. Impaired cell proliferation caused by the expression of the phosphomimetic S20D mutant correlated with the expression of the non-phosphorylatable T21A. These results strongly suggest that the phosphorylation/dephosphorylation of serine 20 and threonine 21 is the determinant that modulates cell proliferation, although we do not exclude the possibility that other types of posttranslational modifications on S20/T21 may control this function. These opposite results suggest that the phosphorylation of T21 may antagonize S20 phosphorylation, likely through conformational changes or steric hindrance. 

The physiologically controlled phosphorylation/dephosphorylation of serine 20 is likely an efficient molecular mechanism for the regulation of the STAU1^55^-mediated posttranscriptional regulation in changing cell environments. Interestingly, STAU1^55^ overexpression had no effect on cell proliferation in non-transformed cells [31]. In contrast, it induced apoptosis in cancer cells. This suggests that cancer cells may express a constitutively active kinase that can phosphorylate STAU1^55^. Cancer cells are known to overexpress several kinases and/or express mutated kinases with constitutively active functions compared to non-transformed cells [49,50]. Since cancer cells grow well, it is likely that only a fraction of STAU1^55^ is phosphorylated and that cancer cells maintain an optimal ratio of phosphorylated/unphosphorylated STAU1^55^ molecules that facilitate proliferation. We proposed that this ratio keeps the cells at the edge between proliferation and apoptosis, since a modest increase in STAU1^55^ expression causes apoptosis. Upon STAU1^55^ overexpression, the absolute amount of phosphorylated STAU1^55^ then increases tipping the balance toward apoptosis. One caveat of this study is that we were unable to document posttranslational modifications on serine 20. Although we identified four phosphorylation sites on STAU1^55^ following immunoprecipitation and mass spectrometry (Boulay and DesGroseillers, unpublished), none of them were in the N-terminal end of STAU1^55^. The absence of coverage in the N-terminal fragment did not allow us to determine if serine 20 is phosphorylated or not and may reflect technical constraints.

### 3.3. Expression of RBD2 Alone Impairs Cell Proliferation and Triggers Apoptosis

Surprisingly, our results indicate that RBD2 is sufficient to impair cell proliferation and trigger apoptosis following its overexpression in transformed cells. As observed with STAU1^55^, the presence of a negative charge on serine 20 of RBD2 was absolutely required to impair cell proliferation. The observation that RBD2 needs the expression of endogenous STAU1^55^ to impair SMD suggests that the charged serine 20 in the context of RBD2 interferes in *trans* with STAU1^55^, the same way as the charged serine 20 in STAU1^55^ does to impair STAU1^55^ posttranscriptional functions. Our model was that the negative charge on serine 20 promotes or facilitates the interaction between RBD2 and the C-terminal domain of endogenous STAU1^55^, preventing STAU1^55^ dimerization and inhibiting SMD (Figure 7). Interactions between RBD2 and RBD5 were previously reported [47,48]. Similarly, the mechanism of STAU1^55^ dimerization was elucidated and shows that the Staufen-swapping domain (SSM) of one STAU1^55^ molecule interacts with the RBD5 of another one [48]. The presence of positively charged amino acids at the SSM-RBD5 interface was consistent with a putative role of the negatively charged serine 20 in a mechanism that regulates STAU1^55^ dimerization. The loss of dimerization would then explain the inhibition of STAU1^55^-mediated posttranscriptional regulation [48]. 

### 3.4. Serine 20 Regulates STAU1^55^ Posttranscriptional Functions

Our results indicate that STAU1^S20D^ loses its ability to enhance translation when bound in the 5′UTR of mRNAs and to induce mRNA decay when bound in the 3′UTR of mRNAs (Figure 5). STAU1^55^-mediated translational regulation also occurs through binding to GC-rich regions of the coding sequence and to the 3′UTR [15,17,18]. STAU1^55^, thus, can strongly influence the expression of RNA regulons in changing cell environment. Many STAU1^55^-bound mRNAs code for proteins involved in the apoptotic or cell proliferation pathways (Appendix A), suggesting that the modulation of STAU1^55^ expression can change the expression of downstream RNA targets and, accordingly, cell fate. Indeed, through ribosome profiling and RNAseq experiments, it was shown that the misregulation of STAU1^55^ expression (overexpression or depletion) changes the translational profiles and/or abundance of multiple mRNAs in transformed cells [15,17,18,31]. Therefore, through impaired posttranscriptional functions, STAU1^S20D^ may completely change the equilibrium between proliferative and apoptotic transcripts and tip the balance toward impaired cell proliferation.

## 4. Materials and Methods

### 4.1. Cell Culture and Transfection

HCT116 (colorectal carcinoma cell line), STAU1-knockout HCT116 [46], and HEK293T (human embryonic kidney cell line) cells were cultured in Dulbecco modified Eagle’s medium (DMEM, Wisent) supplemented with 10% fetal bovine serum (Wisent), 100 μg/mL streptomycin, and 100 units/mL penicillin (Wisent Inc, St-Bruno, QC, Canada) under 5% CO_2_ atmosphere. Cells were transfected with TransIT-LT1 transfection reagent (MJS BioLynx Inc, Brockville, ON, Canada), using 3 µg of plasmid DNA for all transfection, except for plasmids coding for RBD2-tagged proteins (2 µg) and YFP (1 µg) in 10-cm petri dishes. Amounts of 2 µg, 1 µg, and 0.5 µg of plasmids were used, respectively, when transfected in 6-cm petri dishes. For the luciferase assay, a mixture of 600 ng of plasmids coding for the specific proteins, 300 ng of the Rluc or SBS-Rluc vector, and 100 ng of empty vector was transfected in 6-cm petri dishes.

### 4.2. Plasmids and Cloning Strategies

Plasmids cloning for STAU1^55^-HA_3_ (in pcDNA3 RSV) and its progressive N-terminal deletion mutants (Δ88, Δ60, Δ46, Δ37, Δ25, Δ17, and Δ7) were previously described [41]. Plasmids cloning for STAU1^55^-YFP (in YFP Topaz) and its mutant of deletion of the first 88 amino acids were described [47]. To obtain the plasmid RBD2 in YFP Topaz, the region corresponding to RBD2 in STAU1^55^ (amino acids 1–88) was PCR amplified using STAU1^55^-YFP as a template. Briefly, RBD2 was amplified with oligos forward 5′-TACCCGAATTCAGTTATAAGCCTGTTGACCCTTAC-3′ and reverse 5′-TACCACCGGTGATTCTCTTCCATTCACCTCCAG-3′. PCR products were digested with endonucleases EcoRI and AgeI (New England BioLabs Ltd., Whitby, ON, Canada) and ligated in YFP-topaz with T4 ligase (Thermofisher scientific, St-Laurent, QC, Canada).

Phosphomimetics and non-phosphorylatable mutants for S20, T21, Y22, and Y24 of STAU1^55^ were created by substitution with aspartic acid or alanine, respectively. Briefly, we performed all-around PCR assays using oligos (Integrated DNA Technologies, Coralville, IA, USA) (Appendix A) and PfuUltra II Fusion HotStart DNA Polymerase (600670) (Agilent, Toronto, ON, Canada). PCR results were digested with DpnI (NEB-R0176S) (New England BioLabs Ltd., Whitby, ON, Canada). Mutations were confirmed by sequencing. Phospho-mutants for S20 in the RBD2-YFP background were obtained using the same strategy and oligos using RBD2-YFP as the PCR template.

### 4.3. Time Lapse Microscopy and Image Analysis

HEK293T cells were transfected with vectors, allowing the overexpression of STAU1^55^-mCherry or mCherry as a control (in pCDNA3.1-CMV). CellEvent™ Caspase-3/7 Green Detection Reagent (Invitrogen™, ST-Laurent, QC, Canada) was added directly after transfection (Figure 1A experiment A) or 24 h after transfection (Figure 1A experiment B) following the manufacturer’s procedure at a final concentration of 2 µM. Time lapse video microscopy with images taken every 10 min for 12, 16, or 36 h was conducted with the spinning disk Axio Observer Z1 (Zeiss, Oberkochen, Germany), equipped with an incubation chamber providing the optimal cell growth. Similarly to experiment A, HEK293T cells were transfected with previously described vectors STAU1^55^-HA_3_, RBD2-HA_3_, Δ88-HA_3_, STAU1^S20A^-HA_3_, or STAU1^S20D^-HA_3_, and time lapse was performed every 10 min for 20 h. HEK293T cells were transfected with STAU1^55^-YFP, STAU1^S20A^-YFP, STAU1^S20D^-YFP, or YFP control, and single time-point image acquisition on living cells at 37 °C under a 5% CO_2_ atmosphere was conducted after 36 h. Images were analyzed with ImageJ 1.52a (National Institutes Health, Bethesda, MD, USA). YFP (green), and mCherry (red) cell area signals were normalized to the total cell area (brightfield) and expressed as a percentage of this ratio.

### 4.4. Antibodies and Reagents

Anti-HA (12CA5) was purchased from Santa Cruz Biotechnology (Dallas, TX, USA) or from Millipore-Sigma (H6908) (Oakville, ON, Canada). The antibody against GFP (11814460001) was purchased from Roche (Oakville, ON, Canada) and used to detect YFP-tagged proteins since the two proteins are identical except for one amino acid and that anti-GFP antibody perfectly recognizes YFP. The antibody against mCherry was purchased from Sigma-Aldrich (AB356482) (Aokville, ON, Canada). Anti-STAU1 was previously described [51]. Anti-β-Actin (A5441) was obtained from Sigma (Aokville, ON, Canada). All primary antibodies were used at 1:1000 dilution. MG132 (C2211) was purchased from Millipore-Sigma (Aokville, ON, Canada) and used at 20 μM for 8 h. DMSO was purchased from Millipore-Sigma (Aokville, ON, Canada).

### 4.5. Western Blot Analysis

Cell extracts were lysed in Laemli buffer (25 mM Tris-Cl pH 7.4, 1% SDS). Proteins were then quantified with the BCA reagent kit (Pierce^TM^ BCA protein assay) (ThermoScientific, St-Laurent, QC, Canada). After adding the bromophenol blue, 10–20 μg of proteins was separated on 10% (HA-tagged proteins) or 15% (YFP-tagged proteins) acrylamide/bis acrylamide (29:1) gels, analyzed by western blotting, and revealed on X-ray films (Fujifilm, Christie Innomed, St-Eustache, QC, Canada) or with the ChemiDoc MP Imaging System (Bio-Rad Laboratories Ltd, St-Laurent, QC, Canada). Quantification analysis was made with ImageJ or ImageLab 6.1 (Bio-Rad Laboratories Ltd, St-Laurent, QC, Canada) software, respectively.

### 4.6. RNA Isolation and RT-qPCR

Cell extracts were homogenized with TRIZOL reagent (Ambion, St-Laurent, QC, Canada). Nucleic acids were extracted with chloroform and precipitated with isopropanol (Bioshop, Burlington, ON, Canada). Pellets were diluted in water and precipitated twice with LiCl 3M. Samples were digested with DNAse using the TURBO DNA-free kit (Ambion, St-Laurent, QC, Canada). Total RNA was quantified, and 1μg of each sample was reverse transcribed using the RevertAid H Minus First Strand cDNA Synthesis kit (Thermo Scientific, St-Laurent, QC, Canada) and oligo(dT). Products were qPCR amplified with the Luna^®^ Universal qPCR Master Mix (New England BioLabs Ltd, Whitby, ON, Canada) and ran on a LightCycler96 (Roche, Oakville, ON, Canada). Amplification of ARF1 mRNA was performed using the oligos 5′-AGGCTGGTACCGGTCCGGAATTC- 3′ and 5′-CTCTGTCATTGCTGTCCACCACG- 3′, and normalization was made by the average gene expression of HPRT and RPL22, amplified using the oligos forward 5′- GCTTTCCTTGGTCAGGCAGAT -3′ and reverse 5′- CTTCGTGGGGTCCTTTTCACC -3′ for the first and forward 5′- TTGCTGTTAGCAACTACGCGCAAC -3′ and reverse 5′- TGGTGACCATCGAAAGGAGCAAGA -3′ for the latter.

### 4.7. Gene Expression Assays

To study STAU1^55^-dependent translation, plasmids coding for STAU1^55^-YFP, STAU1^S20A^-YFP, STAU1^S20D^-YFP, RBD2-YFP, RBD2^S20A^-YFP, and RBD2^S20D^-YFP were co-transected with plasmids coding for either Rluc or SBS-Rluc [22] in HEK293T cells that were allowed to growth for 24 h. Cells were lysed with passive lysis buffer, and the expression of Rluc was quantified in triplicate using the Dual Luciferase Reporter Assay System (Promega, Madison, WI, USA) and a luminometer (HIDEX). To study SMD, plasmids coding for STAU1^55^-HA_3_, STAU1^S20A^-HA_3_, STAU1^S20D^-HA_3_, and RBD2-HA_3_ were transfected in STAU1-KO HCT116 cells [46]. At 24 h after transfection, the total RNA was extracted and the level of ARF1 mRNA was measured by RT-qPCR, as previously described [23].

### 4.8. Growth Curve Assays

At 24 h post-transfection, cells were trypsinized and plated at the same density (day = 0), and the remaining cells were lyzed and used for western blotting. For growth curve assays, cells were harvested every one or two days, and the number of cells was counted with a hemacytometer (Biorad Laboratories Ltd., St-Laurent, QC, Canada).

## 5. Conclusions

We propose that, following transfection, a cancer-associated kinase phosphorylates STAU1^55^ serine 20 (Figure 7). Phosphorylation of serine 20 impairs STAU1^55^ posttranscriptional activities (translation, mRNA decay) by preventing STAU1^55^ homodimerization or by sequestering factors essential for cell proliferation. The inhibition of STAU1^55^ posttranscriptional activities then changes the expression of STAU1^55^-bound pro/anti-proliferative and pro/anti-apoptotic transcripts and tips the balance toward apoptosis and cell proliferation impairment. STAU1^55^ can use multiple pathways to induce apoptosis, including the activation of the PERK-CHOP pathway of the unfolded protein response [44] and the perturbation of stress granule assembly [51,52,53]. STAU1^55^ is a sensor of cell proliferation and, through modulate expression, controls cell fate. Our results indicate that, in cancer cells, STAU1^55^ is involved in the control of proliferation and apoptosis. Therefore, STAU1^55^ may be considered as a novel therapeutic target against cancer.

## Figures and Tables

**Figure 1 ijms-23-07344-f001:**
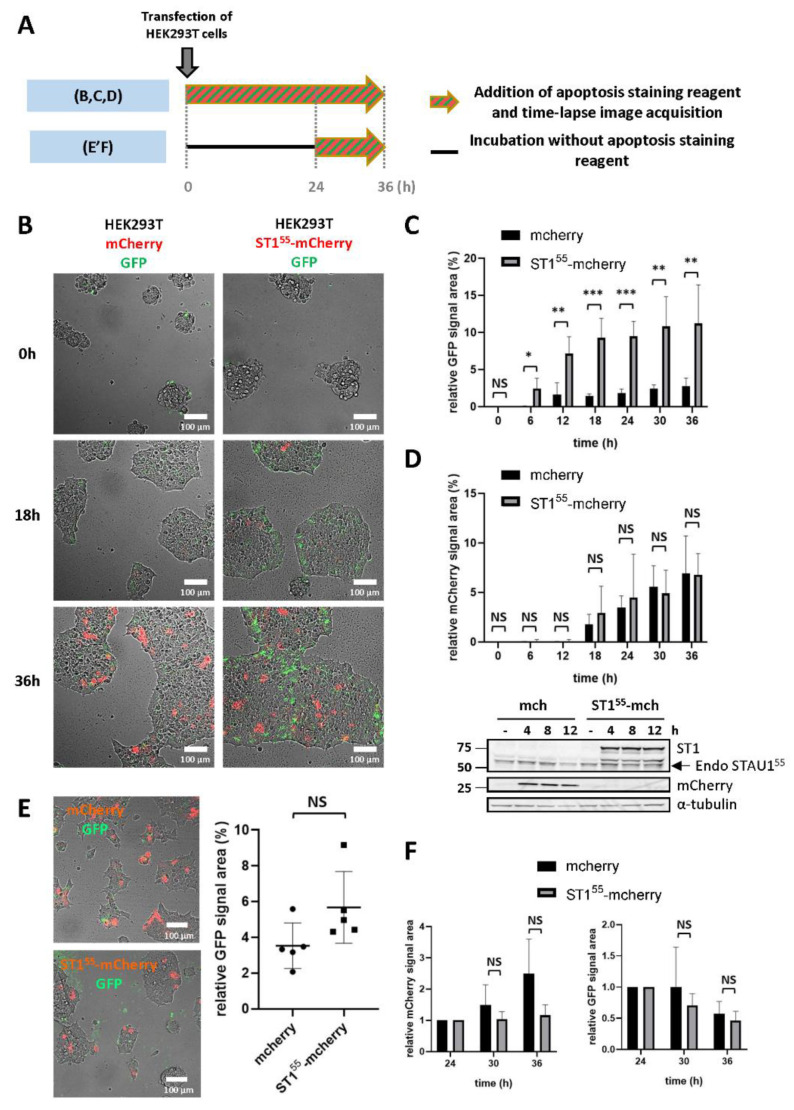
**STAU1 expression induces apoptosis in transformed cells.** (**A**) Schematic representation of the protocols used in the experiments shown in (**B**–**F**). (**B**–**D**) HEK293T cells were transfected with plasmids coding for mCherry or STAU1^55^-mCherry. Caspase activation (apoptosis) was quantified using the CellEvent-caspase3/7-green kit. Following transfection, time lapse video microscopy (**B**) was used to quantify apoptosis (**C**) and mCherry expression (**D**) over 36 h. *, *p*-value ≤ 0.05; **, *p*-value ≤ 0.01; ***, *p*-value ≤ 0.001. NS, not significant. A western blot showed that STAU1^55^-mCherry and mCherry were expressed soon after transfection (**D**). Endo STAU1^55^, endogenous STAU1^55^. (**E**,**F**) At 24 h post-transfection, cells were washed to remove dead cells and incubated in the presence of the CellEvent-caspase3/7-green kit for an additional 12 h. (**E**) Pictures taken 24 h post-transfection showing mCherry-expressing and apoptotic cells (**left**) and the quantification of apoptotic cells (**right**). (**F**) The quantification of apoptosis and mCherry expression at 36 h.

**Figure 2 ijms-23-07344-f002:**
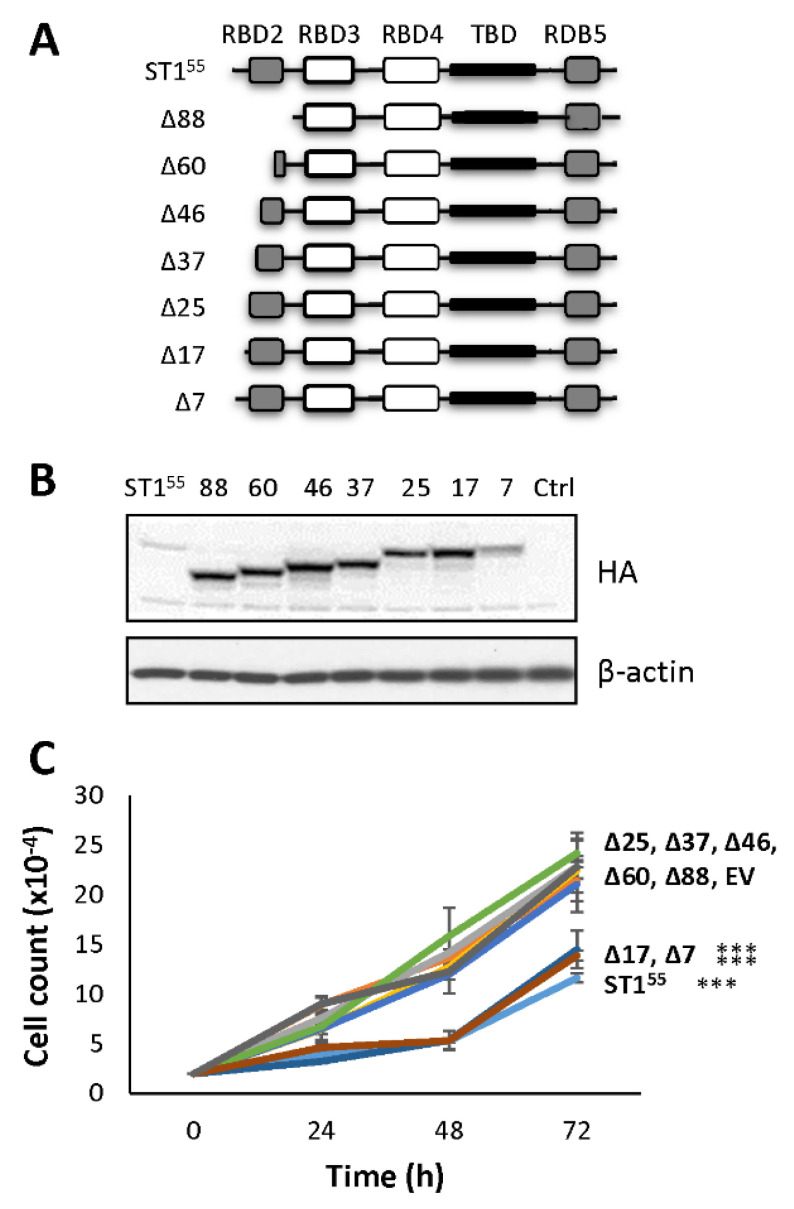
**A molecular determinant that controls cell proliferation is present within amino acids 18 and 25.** (**A**) Schematic representation of STAU1^55^ and STAU1^55^ mutants with progressive deletion in the first 88 amino acids at the N-terminal end of STAU1^55^. RBD, RNA-binding domain; TBD, tubulin-binding domain. White boxes, domain with RNA-binding activity; grey boxes, RNA-binding consensus sequence lacking RNA-binding activity; black boxes, tubulin-binding domain. Δx, deletion of x amino acids at the N-terminal end of STAU1^55^. (**B**) Plasmids coding for STAU1^55^ and STAU1^55^ mutants were transfected in HEK293T cells. At 24 h post-transfection, cells were trypsinized, plated at the same density, and allowed to grow for three days. Expression of the proteins was visualized by western blotting. The blots are representative of three independent experiments that gave similar results. (**C**) Cell proliferation using growth curve assays was quantified over three days. The graph represents the means and standard deviations of three independently performed experiments. ***, *p*-value ≤ 0.001 (Tukey’s multiple comparisons ANOVA test).

**Figure 3 ijms-23-07344-f003:**
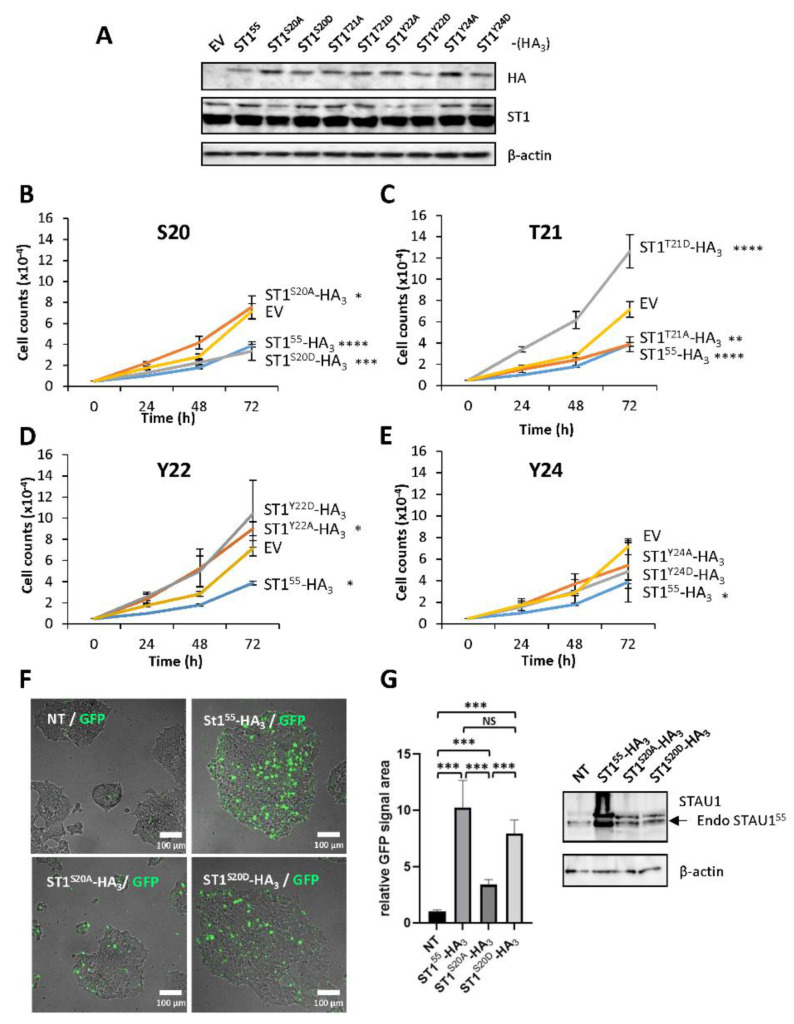
**Serine 20 and threonine 21 controls cell proliferation and apoptosis.** (**A**) HEK293T cells were transfected with plasmids coding for STAU1^55^-HA_3_ or phosphomimetic (S/T/Y to D) and non-phosphorylatable (S/T/Y to A) mutants. The expression of the proteins was visualized by western blotting. The blot is representative of three independently performed experiments that gave similar results. (**B**–**D**) Cell proliferation using growth curve assays to monitor the effect of phosphomimetic and non-phosphorylatable mutations on serine 20 (S20) (**B**), threonine 21 (T21) (**C**), tyrosine 22 (Y22) (**D**), and tyrosine 24 (Y24) (**E**). Transfected cells were trypsinized 24 h post-transfection, plated, and allowed to grow for three days. Each graph represents the means and standard deviations of three independently performed experiments. *, *p*-value ≤ 0.05; **, *p*-value ≤ 0.01; ***, *p*-value ≤ 0.001; ****, *p*-value ≤ 0.0001 (Tukey’s multiple comparisons ANOVA test). (**F**) HEK293T cells were transfected with plasmids coding for STAU1^55^-HA_3_, STAU1^S20A^-HA_3_, and STAU1^S20D^-HA_3_ and immediately incubated in the presence of the GFP-coupled apoptosis stain. (**G**) (Left) Quantification of the GFP signal observed in (**F**). The graph represents the means and standard deviations of three independently performed experiments. ***, *p*-value ≤ 0.001 (Student *t*-test). NT, not transfected. (Right) Western blots showing the expression of the proteins used in (**F**). The blots are representative of three independent experiments that gave similar results. Endo STAU1^55^, endogenous STAU1^55^.

**Figure 4 ijms-23-07344-f004:**
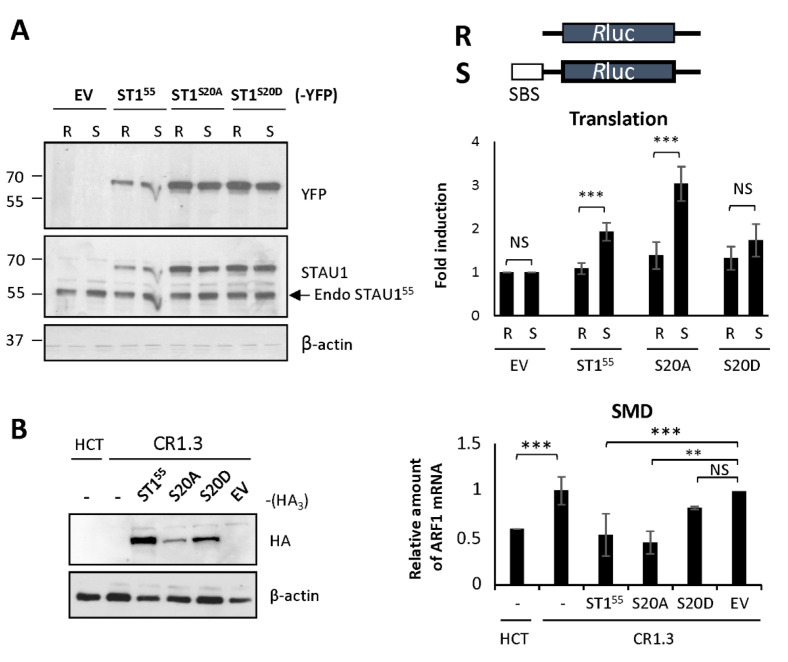
**Expression of the phosphomimetic mutant S20D abrogates** STAU1^55^**-dependent translation and SMD.** (**A**) Schematic representation of the reporter proteins used in the translation assay. Rluc, Renilla luciferase; SBS, STAU1-binding site. HEK293T cells were co-transfected with plasmids coding for the reporter Rluc or SBS-Rluc proteins and for STAU1^55^ and mutants as indicated. (Left panel) Cell extracts were collected 24 h post-transfection, and the expression of STAU1^55^ proteins was analyzed by western blotting. Endo STAU1^55^, endogenous STAU1^55^. (Right panel) Relative translation of Rluc (R) and SBS-Rluc (S) is shown. The graph represents the means and standard deviations of three independently performed experiments. **, *p*-value ≤ 0.01; ***, *p*-value ≤ 0.001. NS, not significant. Translation of Rluc in the presence of the empty vector (EV) was arbitrary fixed to 1. (**B**) STAU1-KO HCT116 (CR1.3) cells were transfected with plasmids coding for STAU1^55^, STAU1^55^ mutants as indicated, or the empty vector (EV). (Left panel) Expression of the proteins was analyzed by western blotting. (Right panel) RNAs were isolated and ARF mRNA was quantified by RT-qPCR using HPRT and RPL22 mRNAs as normalization controls. The graph represents the means and standard deviations of three independently performed experiments. **, *p*-value ≤ 0.01; ***, *p*-value ≤ 0.001. Expression of ARF mRNA in untransfected CR1.3 cells (-) was arbitrary fixed to 1. Untransfected HCT116 cells (-) were used as a reference for SMD.

**Figure 5 ijms-23-07344-f005:**
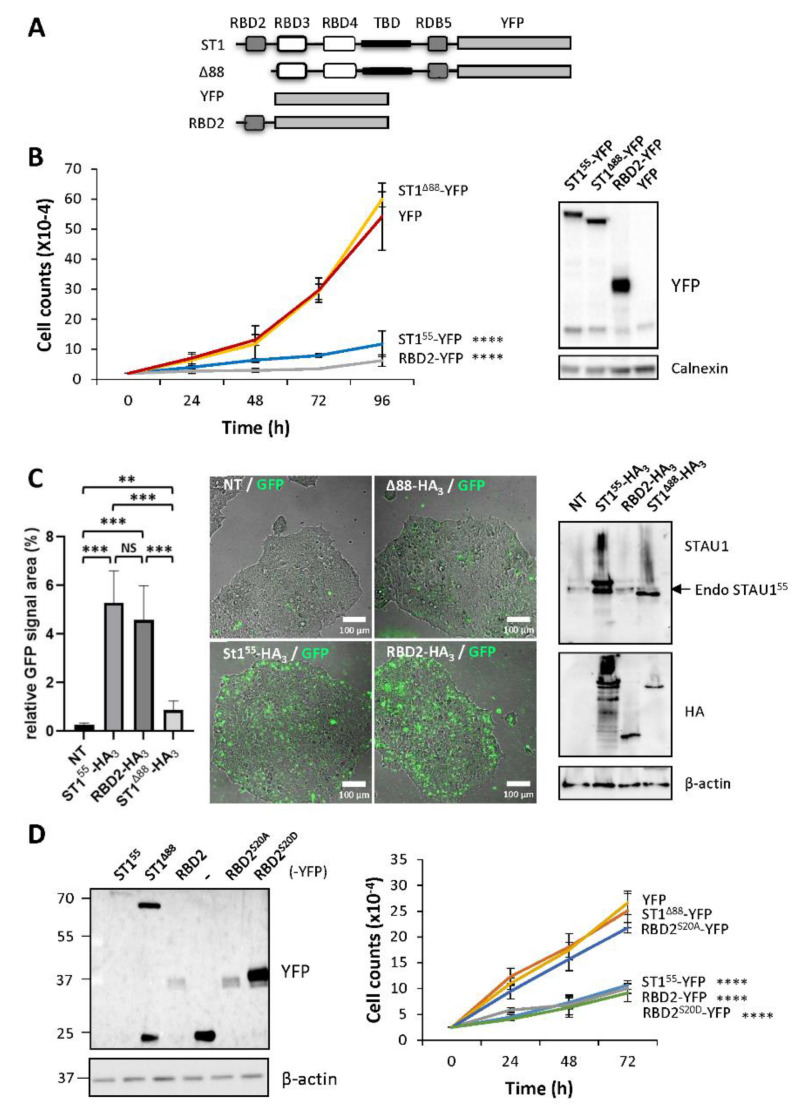
**RBD2 expression impairs cell proliferation via serine 20.** (**A**) Schematic representation of the expressed proteins. See legend of Figure 2. YFP, yellow fluorescent protein. (**B**) Cell proliferation assay. Transfected cells were trypsinized 24 h post-transfection, plated, and allowed to grow for four days. The graph represents the means and standard deviations of three independently performed experiments. ****, *p*-value ≤ 0.0001 (Tukey’s multiple comparisons ANOVA test). Western blots (right panel) showing the expression of the proteins. (**C**) HEK293T cells were transfected with plasmids coding for STAU1^55^-HA_3_, STAU1^Δ88^-HA_3_, and RBD2-HA_3_ and were immediately incubated in the presence of the GFP-coupled apoptosis stain. (**Left**) Quantification of the GFP signal. The graph represents the means and standard deviations of three independently performed experiments. **, *p*-value ≤ 0.01. ***, *p*-value ≤ 0.001 (Student *t*-test). NS, not significant. (**Right**) Western blots showing the expression of the proteins. The blots are representative of three independent experiments that gave similar results. Endo STAU1^55^, endogenous STAU1^55^. (**D**) Cell proliferation using growth curve assays was quantified over three days. The graph represents the means and standard deviations of three independently performed experiments. ****, *p*-value ≤ 0.0001 (Tukey’s multiple comparisons ANOVA test). Western blots (left panel) showing the expression of the proteins.

**Figure 6 ijms-23-07344-f006:**
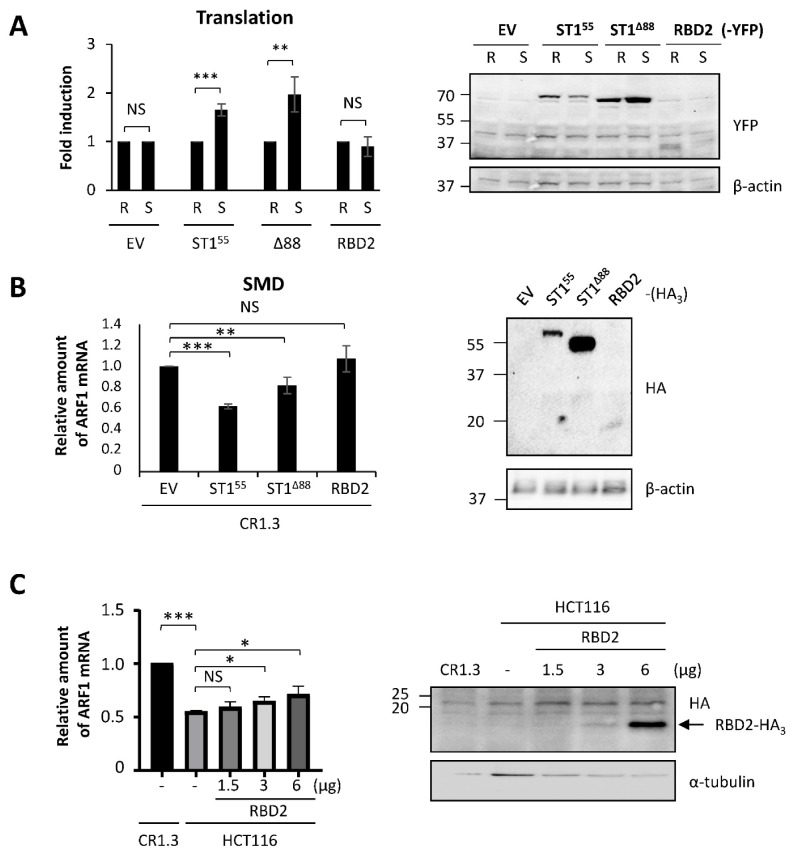
**RBD2 expression impairs SMD via interference with endogenous STAU1^55^ functions.** (**A**) HEK293T cells were co-transfected with plasmids coding for the reporter Rluc or SBS-Rluc proteins and for RBD2-YFP or controls (ST1^55^, Δ88) as indicated. (Left panel) Relative translation of Rluc (R) and SBS-Rluc (S) is shown. The graph represents the means and standard deviations of three independently performed experiments. **, *p*-value ≤ 0.01; ***, *p*-value ≤ 0.001. NS, not significant. Translation of Rluc in the presence of the empty vector (EV) was arbitrary fixed to 1. (Right panel) Cell extracts were collected 24 h post-transfection, and the expression of STAU1^55^ proteins was analyzed by western blotting. (**B**) STAU1-KO HCT116 (CR1.3) cells were transfected with plasmids coding for RBD2-HA_3_ or controls (EV, ST1^55^, Δ88) as indicated. (Left panel) RNAs were isolated, and ARF mRNA was quantified by RT-qPCR using HPRT and RPL22 mRNAs as normalization controls. The graph represents the means and standard deviations of three independently performed experiments. **, *p*-value ≤ 0.01; ***, *p*-value ≤ 0.001. The expression of ARF mRNA in CR1.3 cells transfected with empty vector (EV) was arbitrary fixed to 1. (Right panel) The expression of the proteins was analyzed by western blotting. (**C**) HCT116 cells were transfected with increasing concentrations of plasmids coding for RBD2-HA_3_ as indicated. (Left panel) RNAs were isolated, and ARF mRNA was quantified as above. *, *p*-value ≤ 0.05. The expression of ARF mRNA in untransfected STAU1-KO HCT116 (CR1.3) cells was arbitrary fixed to 1. (Right panel) The expression of the proteins was analyzed by western blotting.

**Figure 7 ijms-23-07344-f007:**
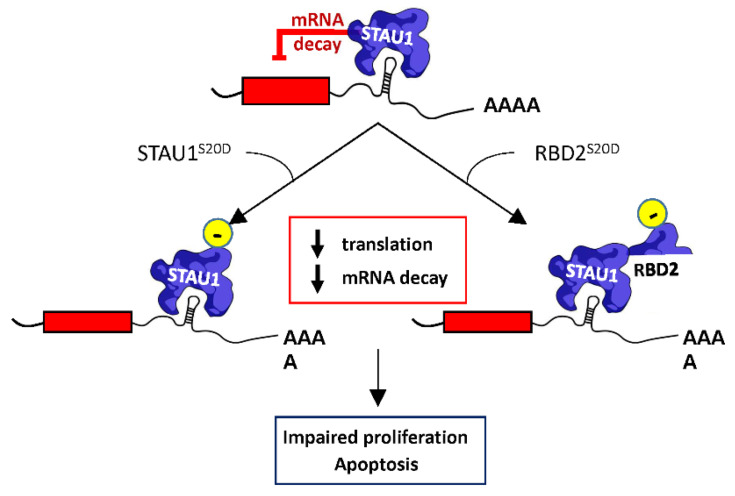
**Proposed mechanism of STAU1^55^-mediated cell proliferation impairment**. STAU1^55^ is a posttranscriptional regulator that controls translation and the SMD of bound mRNAs. Expression of STAU1^S20D^ inhibits both STAU1-mediated translation and decay. Similarly, the expression of RBD2^S20D^ abrogates SMD, but the expression of endogenous STAU1^55^ is required for this phenotype, indicating that RBD2^S20D^ interferes with endogenous STAU1^55^ functions. As a consequence of the misregulation of STAU1^55^-bound mRNAs following STAU1^S20D^ or RBD2^S20D^ expression, cells enter apoptosis or show impaired cell proliferation.

## Data Availability

The data presented in this study are available in the article and Appendix A.

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
