# Peer review of "Phosphomimicry on STAU1 Serine 20 Impairs STAU1 Posttranscriptional Functions and Induces Apoptosis in Human Transformed Cells"

_ijms, 2022, doi:10.3390/ijms23137344_

Round 1

Reviewer 1 Report

The authors provide a convincing study showing the importance of a negative charge (likely due to phosphorylation in vivo) on Ser20 of STAU1. This charge interferes with both its apoptosis inducing and posttranscriptional function. In addition, they show that RBD2 alone with S20D is able to promote apoptosis in the same way as the full length STAU1 S20D. In the end, the authors propose a theory of the reasons why both full length STAU1 and RBD2 both cause apoptosis and issues with RNA decay with S20D. I do hope the authors plan to pursue work on phosphorylation of Y21, as well, because I wanted to know more about that. I realize it would be too large of a paper to do both, however.

Much of the paper is presented very well, but some clarifications are needed. For example, I have no idea what the "55" subscript behind STAU1 is. In the last paragraph of the introduction, the authors go from referring to the protein as STAU1 to referring to it as STAU155 and I don't know why or what the significance is. It is obviously not the first 55 amino acids as they later remove the N-terminal 88 amino acids. So some clarification of that change is needed. Also, what is "Endo STAU155"? It's labeled on some Western blots but not otherwise mentioned.

In general, there is more detail needed about the Western blots. In the methods, please add information on the type of gel used, including acrylamide percentage or percentages. In some of the figures, the western blots don't make sense. If I look at figure 3, I interpret the -[HA3] at the top right of part A to mean that these are HA tagged proteins. It therefore follows that you performed an HA blot. But if you look at Figure 4B, it seems to indicate you did a GFP blot. But from the text and figure 4A and the first part of B, these are YFP proteins, not GFP. Again, in part D, it appears to be a GFP blot, which doesn't make sense. Fixing the figure text and also referring to them in the legend as "anti-YFP western blot" would go a long way to helping the reader make sense of the figure. Figure 5A suffers the same fate - labeled YFP at the top and labeled GFP on the side. The blot in 6A is really confusing because the figure legend and figure refer to YFP fusions, the right side of 6A is labeled YFP on top but GFP on the side, and then the test (lines 297-298) say they are GFP fusions. This inconsistency has to be fixed. In addition, most of the western blot trends do match what the authors state in the results, but they don't seem to mention that you often can barely see the full length protein on the blots at all. The most striking example is in Figure 2B. The full length protein expression level looks to be almost nothing. At least the presence of strong delta17 expression still allows their theory to hold true in this figure. Lastly for the blots, in figure 6B, there's an asterisk that's supposed to refer to non-specific bands but I don't see that asterisk on the figure itself.

The authors' discussion and conclusions are good. With the issues above about the western blots rectified, I think it will be a great paper. Lastly, a couple of language issues. I've never heard the word "signalisation" used in this context (or at all, to be honest). I typically see the word signaling instead. Also, in the figure legends for figures 3 and 4, it refers to cells being trypsinized and plates, rather than plated (it is correct in the figure 2 legend).

Reviewer 2 Report

Quesada et al. present a manuscript on the cellular functions of the double-stranded RNA-binding protein Staufen 1 (STAU1) and characterize in depth the molecular underpinnings behind its functions.

A major caveat of the study is that most experiments were performed in a non-cancer cell line (HEK293T cell line). Although transformed, this is a human embryonic kidney cell line, which has little relevance to the point the Authors want to make about STAU1 as a putative therapeutic target against cancer. Only the STAU1-mediated RNA decay (SMD) experiments were done using the HCT116 colorectal cancer cell line and a HCT116 STAU1-KO. Even though essentially mechanistic, it would be relevant to see some of the key experiments performed in a true cancer context (cancer cell lines), particularly the ones linking the molecular determinant with cellular phenotypes.

The Authors pinpoint the molecular determinant of impaired proliferation and increased apoptosis to the region between amino acids 18-25 of STAU1, namely the S20 residue. The corollary from the experiments is that the endogenous STAU1 is negatively charged at S20 (not shown) and that this post-translational modification prevents STAU1 homodimerization, thus impairing translation and decay of target mRNAs and changing the balance concerning cellular phenotypes. However, transfection with STAU1 55 full-protein (supposedly phosphorylated at S20 and impairing proliferation and inducing apoptosis) increases translation and SMD, while the S20D phosphomimetic mutant (which also impairs proliferation and induces apoptosis) does not (Fig. 5). These results are in direct contradiction with each other and the Authors hypothesis, and thus requires further explanation and discussion.

Another unclear point is the role of the RBD2 domain. The Authors claim that the RBD2 domain can interfere in trans with STAU1 to impair SMD by preventing STAU1 homodimerization. But this is only shown to occur with non-physiological amounts of RBD2 being overexpressed (an artificial condition, as the domain in itself will not appear isolated in the cell). Even if we assume such competition occurs at some degree endogenously, it does not seem to affect STAU1 function, as the expression of the full STAU1 protein does have functional effects in both translation and SMD. So the relevance of this putative regulatory mechanism is unclear.

Specific comments now follow:

Major points

- Fig. 1: The early effect of STAU1 overexpression concerning apoptosis is intriguing, given that cellular apoptosis (GFP+ cells) is already observed before detection of the mCherry reporter signal. An important control is missing: the direct quantification of STAU1 protein expression levels to check for protein expression dynamics, that is to confirm that STAU1 protein is indeed being expressed before detection of mCherry signal and see how it correlates with the GFP signal;

- Results, lines 106 to 110: “The stabilization of the apoptotic GFP signal suggests that once cell fate is decided, STAU155-mCherry-expressing cells no longer enter apoptosis. To test this hypothesis, addition of the GFP-dye and time-lapse microscopy were initiated 24 h post-transfection (Fig. 1A), during the apoptotic plateau (Fig. 1C).” Caspase activation is an early biochemical event in the apoptotic process, hence the assay used probably only detects early apoptosis. If the hypothesis raised by the Authors is correct, could the absence of a significant difference between the two cell lines when the assay is run later be explained by this technical detail? In line with this remark, the GFP signal seems quite low when the dye is added 24h after transfection, compared to when it is added at the same time as the transfection. Please discuss;

- Results, lines 114 to 120: This part is repeated in section 2.2, where the result concerning proliferation is referred in the context of the different mutant constructs. Mentioning it before presenting the remaining figure does not make much sense. It should be removed from section 2.1 for clarity and the relevant parts included in section 2.2;

- Results, line 172: “…presence of a permanent negative charge on S20 affects cell proliferation”. An S20E mutation should also be added to sustain this claim;

- Did the Authors check the effect of threonine 21 modification in apoptosis?;

- Results, section 2.5: The relevance of addressing the function of RBD2 is not clear, as the S20 residue is contained in this domain and mutant constructs with 25 to 88 deleted amino acids (all contained within RBD2) had no effect on cellular proliferation. Furthermore, the RBD2 non-phosphorylatable S20A mutant grows normally. Did the Authors check if the sequence between amino acids 18 and 25 when fused with YFP is enough to cause the phenotypes observed? Please discuss;

- Section 2.6 should appear before section 2.5, so that the two figures addressing RBD2 follow together;

- Fig. 5: STAU1 55 enhances translation of SBS-Rluc, but the phosphomimetic mutant S20D does not. S20D also does not cause SMD, contrary to STAU1 55. How do the Authors reconcile these observations? Please discuss;

- Results, lines 260 to 262: “Luciferase assays first indicated, as expected, that expression of STAU1 enhanced translation of SBS-Rluc compared to Rluc (Fig 5A).”. Why was this result expected? Please explain;

- Results, lines 316 and 317: “…these results indicate that RBD2 acts in trans and interferes with endogenous STAU1 functions to prevent SMD.”. The result shown in Fig. 6C concerning RBD2 competition with endogenous STAU1 is not very convincing, as an effect is only observed with very high concentrations of RBD2-plasmid used. An additional experiment must be performed to sustain this claim. For instance i) RBD2 could be mutated to prevent interaction and check the effect on translation or SMD; or ii) increasing concentrations of STAU1 55 could be co-transfected with increasing concentrations of RBD2-HA3 in HCT116 CR1.3 cells in a competition assay to see if the effect is progressively abrogated; or iii) RBD5 could be mutated (alteration of the positively charged amino acids) to prevent interaction with RBD2 and allow homodimerization;

Discussion, lines 380 and 381: “Our results are consistent with the possibility that the phosphorylation of T21 antagonizes S20 phosphorylation through conformational changes or steric hindrance.”. Please explain which results specifically sustain this claim. Co-transfection experiments with both modifications simultaneously present are not shown to address molecular hierarchy;

Discussion, lines 389 to 393: “One caveat of this study is that we were unable to document posttranslational modifications on serine 20. Although we have identified four phosphorylation sites on STAU1 following immunoprecipitation and mass spectrometry (Boulay and DesGroseillers, unpublished), none of them were in the N-terminal end of STAU1.”. This means no phosphorylated serine 20 residues were detected in mass spec data. Could the Authors elaborate more? Was this analysis done in the overexpression context? Which cell line was used (HEK293T or HCT116)? How to reconcile this result with the cancer-modified kinase hypothesis raised?

Minor points

- For clarity, please explain the nomenclature used for STAU1 protein (STAU1 55) the first time it is mentioned;

- The amount of vector used for most transfections is not indicated in the Figure legends or in the Materials and Methods section. Please, add the missing information;

- Section 2.4 could be included in section 2.3;

- Did the Authors check for the effect of RBD2S20A-YFP and RBD2S20D-YFP in apoptosis? This could be included in Fig. 4;

- Please proofread the manuscript, there are some typos and grammatical errors that need correction.
